# Effective lifting of the topological protection of quantum spin Hall edge states by edge coupling

R. Stühler [1✉], A. Kowalewski[1], F. Reis [1], D. Jungblut[2], F. Dominguez[2,3], B. Scharf[2], G. Li [2,4,5], J. Schäfer [1], E. M. Hankiewicz[2] & R. Claessen [1✉]

The scientific interest in two-dimensional topological insulators (2D TIs) is currently shifting from a more fundamental perspective to the exploration and design of novel functionalities. Key concepts for the use of 2D TIs in spintronics are based on the topological protection and spin-momentum locking of their helical edge states. In this study we present experimental evidence that topological protection can be (partially) lifted by pairwise coupling of 2D TI edges in close proximity. Using direct wave function mapping via scanning tunneling microscopy/spectroscopy (STM/STS) we compare isolated and coupled topological edges in the 2D TI bismuthene. The latter situation is realized by natural lattice line defects and reveals distinct quasi-particle interference (QPI) patterns, identified as electronic Fabry-Pérot resonator modes. In contrast, free edges show no sign of any single-particle backscattering. These results pave the way for novel device concepts based on active control of topological protection through inter-edge hybridization for, e.g., electronic Fabry-Pérot interferometry.

[1] Physikalisches Institut and Würzburg-Dresden Cluster of Excellence ct.qmat, Universität Würzburg, D-97074 Würzburg, Germany. [2] Institut für Theoretische Physik und Astrophysik and Würzburg-Dresden Cluster of Excellence ct.qmat, Universität Würzburg, D-97074 Würzburg, Germany. [3] Institute for Mathematical Physics, TU Braunschweig, 38106 Braunschweig, Germany. [4] School of Physical Science and Technology, ShanghaiTech University, 201210 Shanghai, China. [5] ShanghaiTech Laboratory for Topological Physics, 200031 Shanghai, China. ✉email: raul.stuehler@physik.uni-wuerzburg.de; claessen@physik.uni-wuerzburg.de

In 2D TIs the bulk-boundary correspondence enforces the existence of metallic gap states confined to the one-dimensional (1D) edges of the material. These edge states are spin-polarized and have their spin rigidly locked to the electron's momentum[1]. As a consequence electrons moving along the edge cannot be backscattered by any non-magnetic defects, as schematically depicted in Fig. 1a. This topological edge state protection lies at the heart of the celebrated quantum spin Hall (QSH) effect[2]. It is obvious that the unique properties of the topological edge states lend themselves to a plethora of novel functionalities and application ideas, ranging from low-power consumption electronics to innovative spintronics devices to possible solid-state realizations of qubits[3].

Most theoretical device concepts are based on direct control of the topological edge states by external stimuli, such as electric or magnetic fields, or by bringing them into spatial proximity to ferromagnetic[4] or superconducting materials. Ideas include in particular various types of field-effect transistors (FET). For example, the symmetry-breaking effect of an electric gate field can be used to trigger a phase transition of the 2D TI to a trivial insulator, as recently demonstrated for $Na_3Bi$[5]. The on/off characteristics of such a FET is determined by the complete quenching of the current-carrying topological edge states[6–8]. In an alternative FET concept relying on much smaller gate fields the Fermi level is toggled between an in-gap position and the bulk band edges. In the former situation the edge states carry a dissipationless current, while in the latter they will couple to the dissipative bulk states, thereby effectively losing their topological protection. The resulting

promotion of backscattering from impurities and phonons is estimated to allow on/off ratios of more than two orders of magnitude[9].

A more direct way of controlling and eventually lifting topological protection is achieved by tunneling between opposite edges of a 2D TI[10–17]. The resulting hybridization between right(left)-moving electrons on one edge with their left(right)-moving partners of like spin on the opposite edge will open a small gap at the Dirac point and–even more importantly–create a channel for electron backscattering without the need to break time-reversal symmetry, as depicted in Fig. 1b. A possible realization of such a situation is a narrow constriction in a 2D TI as, e.g., engineered by nanopatterning or defined by suitable line defects in the atomic lattice[18]. Interedge tunneling will then lead to the formation of Fabry-Pérot-type resonances along the constriction and hence a modulated transmission through the device, controlled by the position of the Fermi level[12]. While numerous theoretical proposals along this line have been put forward, there exist surprisingly few experimental studies on edge coupling in a QSH insulator. Strunz et al.[19] have recently studied the effect of Coulomb interaction between 2D TI edges in spatial proximity, whereas Jung et al.[20] focus on the tunneling-induced gap opening in the 1D edge states of a topological crystalline insulator.

## Results and discussion
In this paper we examine the effect of topological edge coupling by spatial mapping of the resulting wave function, thereby

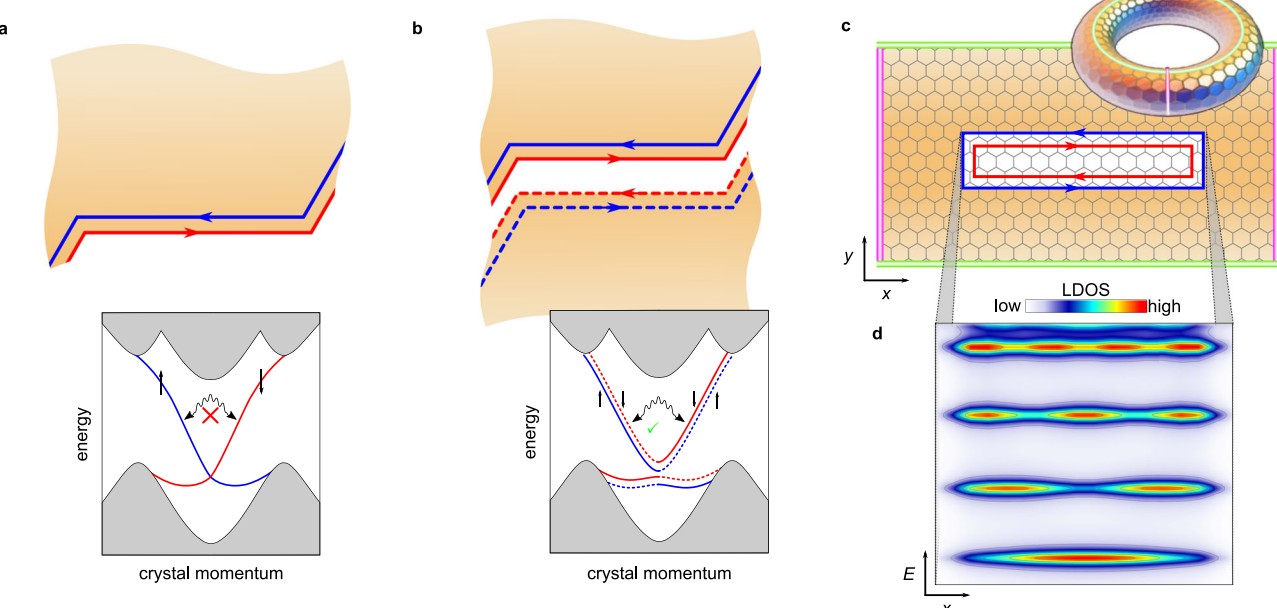

**Fig. 1 Free vs. coupled helical edge states. a** Edge segment of a generic 2D TI material. A pair of helical edge states is bound to the edge. Back-scattering from defects such as the edge kinks is impeded by topologically protected spin-momentum locking, i.e., left-moving spin-up states (blue) cannot scatter with right-moving spin-down states (red). The corresponding one-dimensional band structure is schematically depicted below, where the arrows represent the electron spin projection and the wavy line indicates (prohibited) single-particle backscattering. **b** Two opposing edge segments in close proximity. Spatial overlap of both pairs of helical edge states induces hybridization (or tunneling) between both edges, thereby allowing inter-edge scattering from one segment to the other across the boundary. Strikingly, this opens a channel for back scattering from the edge kinks, i.e., the topological protection becomes partially lifted due to inter-edge hybridization. The corresponding band structure and the now allowed single-particle backscattering is depicted below. **c** Tight-binding model of coupled edge states in the 2D TI bismuthene. The white area indicates a topologically trivial line defect along the x-direction embedded in the topological bulk (light brown). The green and pink boundaries of the finite-size bulk are connected via periodic boundary conditions, resulting in the depicted torus. The trivial line defect connects two opposite edges; the resulting wave function overlap between both sides induces hybridization of like-spin edge states across the line defect (y-direction) as in **b**. **d** Energy-dependent LDOS of the tight-binding model along the line defect in **c** (integrated across its y-width). The line defect induces standing wave excitations bound to its longitudinal extent (along x) and, as the underlying helical states of bismuthene exhibit an approximate linear dispersion, are linearly quantized in energy E. For details of the calculation see Supplementary Sec. F.

revealing direct experimental evidence for the loss of topological protection without breaking time-reversal symmetry. For this purpose we use bismuthene as prototypical 2D TI[21], in which narrow constrictions appear naturally in the form of line defects. Bismuthene consists of a honeycomb monolayer of Bi atoms covalently bonded to a SiC(0001) substrate. Its band structure features a large topological band gap of ~0.8 eV[22], inducing helical edge states which due to their narrow spatial confinement of only a few Angstroms are ideally suited for wave function mapping by STM/STS[21,23].

**Theoretical model.** First insight into the physics of edge coupling in bismuthene can be obtained from a tight-binding (TB) approach based on first-principles calculations[21,24] (see Supplementary Sec. F. for details). In our model we place a line defect into the bismuthene bulk, consisting of a stripe-like region of topologically trivial material, as depicted in Fig. 1c. Effectively this can be thought of as two finite-length edges of the topological bulk coupled across a narrow trivial gap. Indeed, we observe the formation of helical edge states along both edges of the line defect which counterpropagate with opposite helicity. Importantly, the line defect is narrow enough to generate a spatial overlap of the edge state wave functions across the defect (i.e., in $y$-direction), resulting in the expected hybridization of like spin states. This in turn allows backscattering at the two terminations of the line defect and subsequently induces the formation of standing waves, similarly to the behavior of light waves in an optical resonator[25,26]. Figure 1d displays the corresponding Fabry-Pérot oscillations in the local density of states (LDOS). The quantized resonance levels are linearly spaced in energy as a direct consequence of the linear dispersion relation of the underlying bismuthene edge states[21]. We note that in the case of bismuthene, the edge state dispersion is highly asymmetric around the Dirac point as derived from DFT calculations[21], with quasi-linear dispersion above the Dirac point, but rather flat behavior below it (as sketched in Fig. 1a, b). Due to this peculiar band situation, Fabry-Pérot-type resonances in bismuthene are expected to evolve only at energies above the Dirac point, which itself is expected to lie energetically below the valence band maximum[21].

**Topography of domain boundaries.** For the experiments, the bismuthene monolayers were prepared by molecular beam epitaxy using a Knudsen cell as Bi source and SiC(0001) as supporting substrate (for details see Ref. [21] and the Method section). Bismuthene forms a $\sqrt{3} \times \sqrt{3}R30°$ superstructure of Bi atoms in planar honeycomb geometry on the SiC(0001) substrate, clearly resolved in the topographic STM overview of Fig. 2a. For such a superlattice three distinct yet equivalent phase registries exist with respect to the substrate lattice, resulting in the formation of phase domains separated by phase-slip domain boundaries (DBs)[27]. In Fig. 2a the DBs appear as meandering line defects and, importantly, display a distinct periodic structure along their piecewise straight sections, see window 1. The DBs are observed only between zigzag terminations of the connected bismuthene domains, indicating that DBs formed by armchair terminations are energetically unfavorable. While the individual DB straight sections in Fig. 2a are relatively short we generally observe a wide distribution of lenghts up to ~25 nm (see Supplementary Fig. S3). In addition to the DBs we also observe free edges with zigzag termination at regions where the film growth is locally incomplete, see window 2 in Fig. 2a for an example. In this case the terminated domain is not directly connected to another domain.

Further insight into the local DB structure can be obtained by direct comparison to the free zigzag edges, using set-point dependent STM constant current measurements. The DBs reveal

characteristic topographic features at certain bias voltages as shown in Fig. 2b: arrow-shaped segments at 1.3V, bone-shaped segments at 1.8V, and a groove at 2.0V. Fig. 2c demonstrates that the DB structure can indeed be understood as the combination of two free zigzag edges: it is composed of bias-dependent STM images of a free zigzag edge, additionally mirrored and stitched together at the appropriate mirror axis. One can clearly identify the same structural features as for the DB in Fig. 2b, though slightly shifted in bias. This remarkable agreement allows a precise identification of the lattice phase slip between the domains 'A' and 'B' joined by the DB, visualized by the red and green honeycomb lattices in Fig. 2b, c. A second result of this comparison is the determination of the separation between the two zigzag edges forming the DB, which amounts to 15Å (see Supplementary Fig. S1). Finally, excellent correspondence is also observed for the line profiles of both DB and free zigzag edge (Fig. 2d). In particular, the profiles reveal a period doubling of DB and free edge ($2a_{Bi} = 10.7$ Å) with respect to the lattice constant $a_{Bi}$ of bulk bismuthene. This behavior is strongly reminiscent of the so-called (57) reconstruction reported for graphene zigzag edges[28–30]. Furthermore, for graphene/Ni(111) (558)-type DBs have been observed which form from two merging (57)-reconstructed zigzag edges[27], in close correspondence to our present findings.

**LDOS mapping of domain boundaries.** Having thus established that the bismuthene DBs can structurally be viewed as coupled zigzag edges we now turn to their electronic properties, again in direct comparison to their free counterparts. Because the helical edge states in bismuthene are confined to within only a few atomic distances from the boundary, their spatial charge distribution is ideally addressed by STM/STS. Specifically, the local differential conductivity ($dI/dV$) recorded by this technique is a direct measure of the LDOS. In a perfect lattice the LDOS is spatially modulated only by the lattice periodicity. However, local imperfections, such as the kinks in our DBs, can give rise to additional modulation of the $dI/dV$ signal through QPI, i.e., the interference between incoming and elastically scattered electron wave. The modulation wavelength is given by $\lambda(E) = \pi/k(E)$, where $E$ and $k$ are energy and momentum of the electron, respectively. QPI is thus a powerful method to probe the presence vs. absence of single particle backscattering, and hence of topological protection.

Experimentally, the existence of metallic edge states in bismuthene has so far only been established for armchair edges induced at substrate terrace steps[21,23]. It remains to be demonstrated that topological edge states also exist at free zigzag edges. Figure 3b depicts such an edge, terminating the bismuthene domain in the upper half of the image. The edge is not a continuously straight line, but consists of several straight segments of variable length, interconnected by kinks where the edge changes direction by multiples of 60°, thereby maintaining the zigzag nature of the edge.

We first measure the $dI/dV$ signal far from the edge in the bismuthene bulk at a position marked by the gray square in Fig. 3b. The corresponding spectrum (gray-shaded curve in Fig. 3a) confirms the expected large band gap. In contrast, the $dI/dV$ spectra measured immediately at the free zigzag edge (positions marked by red and green squares in Fig. 3b) consistently show a filling of the entire bulk gap with spectral weight (red and green spectra in Fig. 3a). The smooth and essentially featureless edge spectra confirm a homogeneous metallic LDOS closely confined to the zigzag edge, in full correspondence to the observations at armchair edges[21]. Note that the V-shaped dip at zero bias is attributed to

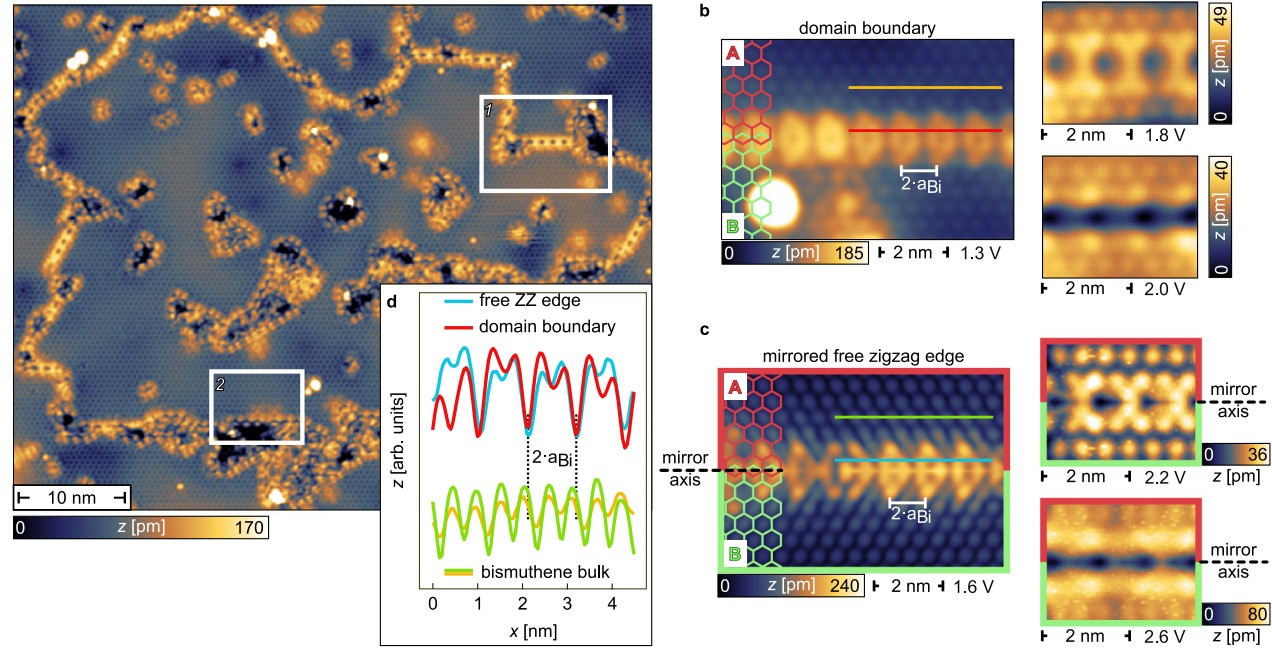

**Fig. 2 Domain boundary and zigzag edge topography. a** STM constant-current map showing a topographic overview of bismuthene. Its $\sqrt{3} \times \sqrt{3}R30°$ superstructure with respect to the SiC(0001) substrate surface results in the formation of three distinct phase domains. The meandering network of line defects are DBs separating the domains from each other. Window 1 marks a particularly well-ordered straight DB section. The map also shows free zigzag edges encircling uncovered substrate surface due to incomplete monolayer growth, see window 2. $V_{set} = -1.0$ V, $I_{set} = 10$ pA. **b** STM topography of a DB recorded at different set-points ($V_{set} = [1.3$ V, $1.8$ V, $2.0$ V]), revealing bias-dependent structural features. The yellow and red lines mark the positions of the corresponding line profiles in **d**. $I_{set} = [50$ pA, $50$ pA, $50$ pA]. **c** STM topography of a free zigzag edge recorded at $V_{set} = [1.6$ V, $2.2$ V, $2.6$ V]. The images show the free zigzag edge mirrored at the marked mirror axis such that domain 'A' is mirrored into domain 'B' to properly generate an adsorption site shift to mimic the situation at a DB. The registry mismatch between both domains can be seen in the respective hexagonal lattices. Green and blue lines mark the positions of the line profiles in **d**. $I_{set} = [70$ pA, $100$ pA, $100$ pA]. The procedure of mirroring is exemplified for one zigzag edge in Supplementary Fig. S5. **d** STM constant-current line profiles from the STM topographies in **b** and **c**, highlighting the similarly doubled periodicities of free zigzag edge and DB with respect to the bismuthene bulk.

Tomonaga–Luttinger liquid physics in the 1D edge state, based on its specific energy- and temperature dependence[23], and should not be confused with a gapped Dirac point. Further LDOS mapping confirms that the metallic edge channel extends along the circumference of this particular bismuthene domain, see Supplementary Fig. S2. The spatial behavior of the energy-dependent LDOS along the white-dashed path in Fig. 3b is shown in Fig. 3c. The only spatial modulation that can be identified in this data is the $2a_{Bi}$-periodicity due to the atomic structure of the zigzag edge. Its purely structural origin is corroborated by its energy independence. No additional and, in particular, energy-dependent modulations are seen that would reflect any QPI. Consequently, we are led to conclude that no backscattering occurs at the kinks between the piecewise straight edge segments, consistent with the topological protection of the helical edge states at the free zigzag edges.

Completely different behavior is observed at the DBs. Figure 3e depicts a DB separating two bismuthene phase domains in the upper and lower half of the image. Like the free zigzag edge, the DB is not straight throughout, but consists of three adjacent straight segments interrupted from each other by kinks (strictly speaking: double-kinks). The local tunneling spectra are measured off and on the DB (positions marked by the colored squares in Fig. 3e) and shown in Fig. 3d. While the bulk spectrum (gray) is practically unchanged with respect to the previous case, the DB spectra (green, blue, and red curves) differ significantly from the smooth gap filling seen for free zigzag edges. They are instead characterized by rapid variations with energy evolving into distinct LDOS peaks near and below zero bias, (labeled $E_1$–$E_4$ in Fig. 3d). The peak intensities display a strong spatial modulation

along the DB, even when measured at topographically equivalent positions (see, e.g., green vs. blue spectrum), pointing towards a non-structural origin. This is even better seen in Fig. 3f showing the full energy-dependent LDOS measured along the DB. The spatial $dI/dV(E, x)$ dependence not only reflects the DB's morphological periodicity of $2a_{Bi}$, but, in stark contrast to the free zigzag edge, features additional periodic modulations at the discrete energies $E_1$ to $E_4$ of the spectral peaks in Fig. 3d. Their wavelength decreases towards higher energy, reminiscent of standing waves in a resonator. This interpretation is further corroborated by the two-dimensional LDOS maps of the DB in Fig. 3g measured at the resonance energies $E_1$ to $E_4$. The modulations are indeed wave-like along the direction of the DB, with their wavelength $\lambda$ directly inferred from the width-integrated $dI/dV$ line profiles also contained in Fig. 3g.

All observations are consistent with the formation of electronic Fabry-Pérot states[26], resulting from QPI due to backscattering off the DB kinks. We note in passing that at higher energies (bias > $E_4$) the QPI wavelength will become comparable to the lattice constant, resulting in complex beating between structural and electronic periodicities. This may account for the complicated $dI/dV$ spectral behavior in this energy region (see Fig. 3d). The Fabry-Perót picture of partially reflected waves also accounts for the observation that the $dI/dV$ data in Fig. 3d, f, g is composed of two components: (1) a smooth background signal, and (2) a strongly modulated (peaky) component along the domain boundary. We associate component (1) with the unscattered, i.e., transmitted part of the propagating edge state through the kink, and component (2) with the backscattering-induced interference effect.

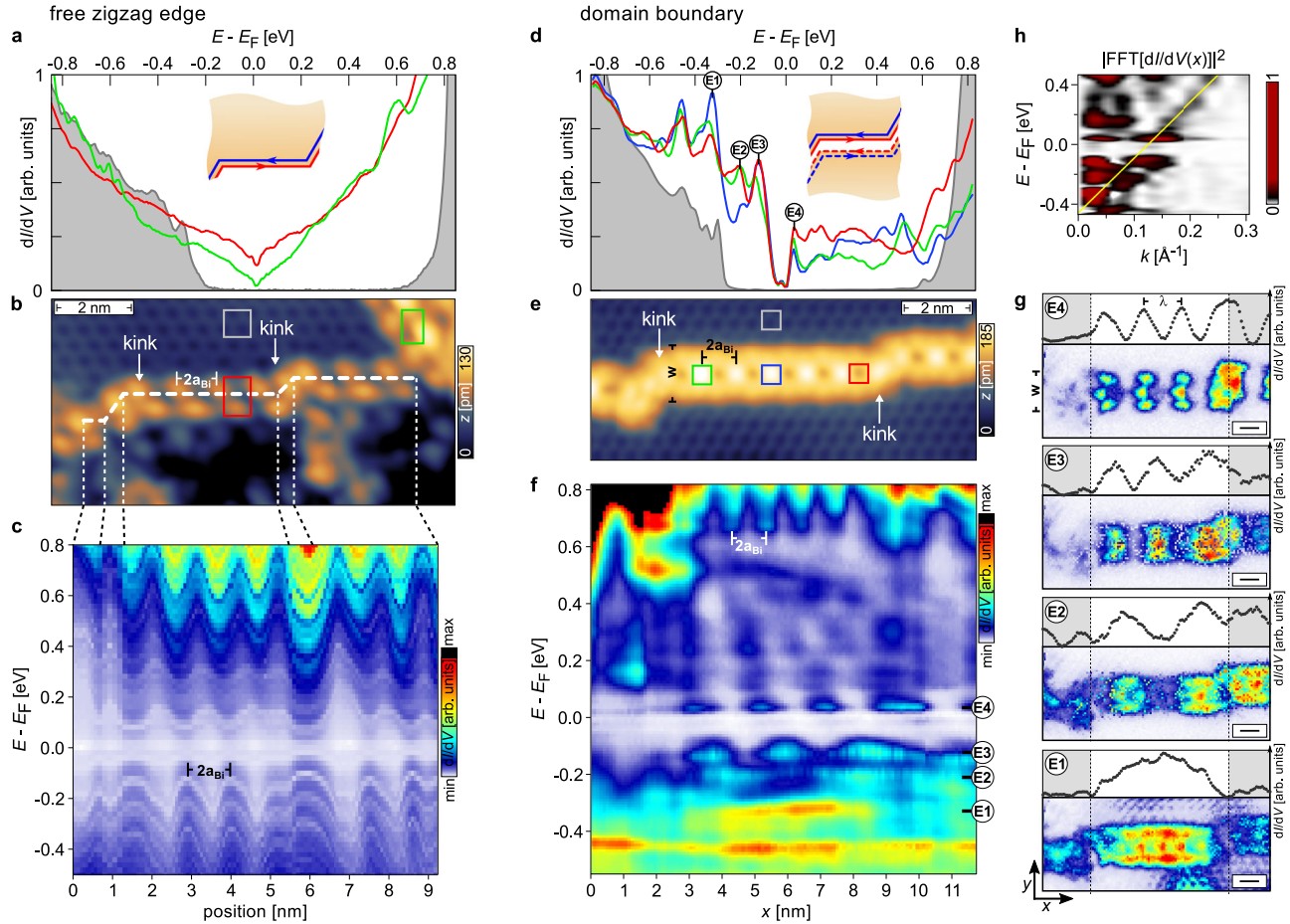

**Fig. 3 Electronic properties of free zigzag edge states vs. DB states. a** STS d$I$/d$V$ spectra on/off a free bismuthene zigzag edge, measured at positions marked in **b**. The bulk spectrum (gray) shows an insulating gap. Spectra taken on the edge (red, green) verify the existence of a metallic edge channel smoothly filling the gap. **b** Constant-current STM topography of a free zigzag edge ($V_{set} = -1.2$ V, $I_{set} = 20$ pA). The edge is divided into straight edge segments of variable length separated by kinks (indicated by white-dashed line). Colored squares: positions, where the spectra in **a** are taken. **c** Energy-dependent d$I$/d$V$ signal versus position along the white-dashed path in **b**. The periodic modulation corresponds to the structural edge periodicity $2a_{Bi}$. **d** d$I$/d$V$ spectra on/off a bismuthene DB, at positions marked in **e**. The bulk spectrum (gray) agrees with the one in **a**. The spectra taken on the DB (green, blue, red) (nearly) fill the gap and are characterized by the presence of intense peaks at discrete energies, in remarkable contrast to the zigzag edge spectra in **a**. **e** STM topography of a DB ($V_{set} = -0.85$ V, $I_{set} = 80$ pA). Similar to the free zigzag edge, the DB consists of piecewise straight sections connected by kinks. Colored squares: positions, where the spectra in **d** are taken. **f** Energy-dependent ($w$-integrated) d$I$/d$V$ signal versus position along the DB in **b**. Energy-dependent standing wave excitations are observed at the marked energies $E_1$ to $E_4$, additional to the structural $2a_{Bi}$ modulation. **g** Two-dimensional d$I$/d$V$ maps of the DB in **e**, taken at the discrete energies $E_1$ to $E_4$ in **f**. The corresponding ($w$-integrated) d$I$/d$V$ profile along the DB is shown above each map. The modulation wavelength $\lambda$ is extracted from the d$I$/d$V$ profiles. Gray-shaded regions: guide to the eye to mark the extension of the effective resonator. The scale bar is 1nm. **h** Power spectrum of the d$I$/d$V$ map in **f**. Yellow line: guide to the eye with a slope of 3.6 eVÅ.

**Linear scaling of Fabry–Pérot resonances.** Our combined data thus provide clear evidence that the protection against back-scattering observed for the free zigzag (and armchair[23]) edge is lifted for the metallic DB states, at least partially. What remains to be shown is that the Fabry-Pérot states in the DBs are composed of coupled topological edge states and not just caused by QPI of trivial boundary states. For this purpose we utilize the fact that the standing wave excitations carry information on the $E(k)$ dispersion of the underlying electronic states. It can be directly inferred from our STS results, with a first hint already given by the d$I$/d$V(E, x)$ data in Fig. 3f. Their Fourier transform into reciprocal space is shown in Fig. 3h where the Fabry-Pérot modulations generate a linear $E(k)$ structure (marked by the yellow line), as expected for a Dirac-like topological edge state. For a more systematic determination of the dispersion we have analyzed the standing wave excitations in the d$I$/d$V$ maps of many different straight and defect free DB sections, with lengths ranging from 3.2nm to 8.6nm. For each DB section we

determined the wavelength $\lambda$ of its standing waves as a function of energy. In order to correct for extrinsic energy shifts due to local variations of the chemical potential (see Supplementary Fig. S4), all energies are referred to the bulk valence band onset $E_{VBM}$, individually measured in the local vicinity of each respective DB. The standing wave energies thus obtained are plotted in Fig. 4a as function of $k = \pi/\lambda$. The data points are consistently described by a linear dispersion of the form

$$E - E_{VB} = E_D + \hbar v_F k, \qquad (1)$$

where $E_D$ is the Dirac point energy relative to $E_{VBM}$ and $v_F$ the Fermi velocity. A possible small hybridization gap due to inter-edge tunneling (cf. Fig. 1b) is not resolved here. A numerical fit of our data with the dispersion of equation (1) yields $E_D = -0.39(8)$ eV and $\hbar v_F = 3.6(6)$ eV Å. For comparison, the theoretical Fermi velocity predicted for the bismuthene zigzag edge is $\hbar v_F = 4.3$ eV Å. Density-functional theory (DFT) predicts for the topological state at a bismuthene zigzag edge a bare Fermi

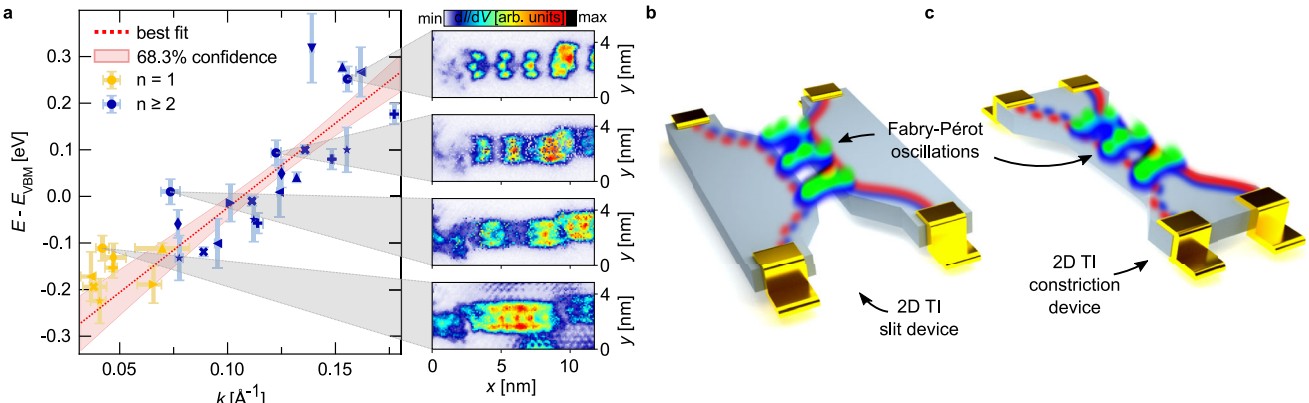

**Fig. 4 Fabry–Pérot interferometry in a 2D TI nanoconstriction. a** Fabry–Pérot resonator modes labeled by index n as a function of $k = \pi/\lambda$. The modes of equal DBs are assigned equal marker symbols. Red dotted line: Least squares fit to the data for $n \geq 2$ (see Methods for details). Red shaded region: 68.3% confidence bands of the fit. The constant energy $dI/dV$ maps on the right-hand side (taken from Fig. 3g) are assigned to the respective data points. The energy error bars reflect the s.d. related to the valence band onset ($E_{VB}$) determination, see Supplementary Fig. S4. The $k$ error bars for $n \geq 2$ relate to the s.d. in determining $\lambda$. The $k$ error bars for $n = 1$ relate to the estimated error in determining the length of the specific DB. **b, c** Schematic four-terminal devices for electronic Fabry–Pérot interferometry based on a QSH insulator in slit or, alternatively, constriction geometry.

velocity of 2.6 eV Å[21]. However, this DFT calculation (using a PBE exchange-correlation potential) strongly underestimates the indirect band gap ($E_{gap}^{DFT} = 0.48$ eV) spanned by the edge state. Its actual experimental value is $E_{gap}^{exp} = 0.8$ eV [21,22]. Renormalizing $\nu_F$ by the factor $E_{gap}^{exp}/E_{gap}^{DFT}$ yields the value given above. This good correspondence provides smoking gun evidence that the Fabry–Pérot states observed in the bismuthene DBs are indeed derived from pairs of topological edge states, and that their coupling opens a channel for single-particle backscattering, i.e., that their topological protection is lifted.

**Related work**. It is interesting to put these results into perspective with respect to related recent work. Pedramrazi et al.[31] have conducted STM/STS on similar DBs in the 2D TI $1T'$-WSe$_2$. While they report qualitative differences in spectral line shape in the DBs versus interfaces between topological and trivial phases, no backscattering-induced QPI could be detected. Further, Howard et al.[32] do observe QPI at step edges of the quantum anomalous Hall material $Co_3Sn_2S_2$ induced by random local impurities. They relate their findings to the coupling of chiral edge states in this material system with broken time-reversal symmetry.

**Conclusion and outlook**. In summary, the intrinsic phase-slip DBs in bismuthene realize the direct coupling between spin-momentum locked edge states on either side of the DB. This results in the loss of their topological protection against single-particle back-scattering, manifesting itself in QPI, i.e., the formation of Fabry–Pérot states. By reversing the argument, the absence of any QPI signatures in the free zigzag edges provides strong experimental confirmation that the metallic edge states in the bismuthene band gap are protected against scattering and hence indeed of topological character. These findings have bearing for quantum transport experiments on bismuthene. If not performed on a single phase domain, the presence of DBs may introduce unwanted dissipation and hence impede the detection of a clean QSH effect. From a broader perspective, our direct wave function mapping of coupled topological edges strongly encourages novel device concepts for 2D TI-based electronic Fabry–Pérot interferometry (FPI). Previous experimental realizations of electronic FPI have their drawbacks, such as lack of scalability when using carbon nanotubes[25], or the need for high

magnetic fields when based on chiral edge states in a quantum Hall device. Instead, if narrow enough a simple slit or constriction nanopatterned into a QSH insulator (see Fig. 4b, c) that is suitable for this lithographic process is sufficient to generate coupling between its helical edge states, similar to the situation in our bismuthene DBs. Additionally, suitable gate electrodes could be used to tune the effective length and edge coupling strength of the constriction and hence provide wide control of its electronic transmission. Side gates have also been suggested to allow separate control of charge and spin transport[11–13]. We are confident that our microscopic study of edge coupling in a QSH insulator will fertilize the realization of such quantum transport experiments.

## Methods

**Bismuthene synthesis**. Bismuthene was grown on n-doped 4H-SiC(0001) substrates (12.0 mm × 2.5 mm) with a resistivity of 0.01 Ω cm − 0.03 Ω cm at room-temperature. The dopant concentration of the substrate is $5 \times 10^{18}$ cm$^{-3}$ − $1 \times 10^{19}$ cm$^{-3}$. The atomically flat and well-ordered surface of the substrate was prepared in a hydrogen dry-etching process[33] by annealing the sample at 1180 °C for 7 min in a gas atmosphere (950 mbar) using 2 slm $H_2$ and 2 slm He both with a purity of 7.0 and additionally filtered using gas purifiers. The hydrogen passivated SiC sample was then transferred into the UHV growth chamber for bismuthene epitaxy. The epitaxial process of bismuthene contains two steps. First, the hydrogen is desorbed from the SiC surface by heating the sample to 600° for ~2 min. Then, pure bismuth (99.9999% purity) was evaporated from a standard Knudsen cell with the sample held ~550 °C for ~2 min.

**STM measurements**. The STM/STS measurements have been performed with a commercial low-temperature STM from Scienta Omicron GmbH under UHV conditions ($p_{base} = 2 \times 10^{-11}$ mbar). Topographic STM images are recorded as constant current images. After stabilizing the tip at a voltage and current set-point $V_{set}$ and $I_{set}$, respectively, the feedback loop is opened and STS spectra are obtained making additional use of a standard lock-in technique with a modulation amplitude between $V_{mod} = 5$ mV and $V_{mod} = 12$ mV. All measurements were performed at liquid He temperature, i.e., $T = 4.35$ K. Because of external modulation ($V_{mod}$) the instrumental resolution results in a convolution with a semicircle ($e * \sqrt{V_{mod}^2 - V^2}$) in addition to the thermal broadening. Prior to every measurement on bismuthene we assured that the tip DOS is metallic by a reference measurement on a Ag(111) surface.

**Fitting procedure**. The linear fit of the energy vs. momentum data in Fig. 4a was performed for data points with level index $n \geq 2$ only. For such modes the wavelength $\lambda$ of the electronic standing waves is easily determined from the separation of two adjacent charge maxima (see Fig. 3g). In contrast, for $n = 1$ modes $\lambda$ has to be estimated from the distance of the outer nodal points, i.e., the longitudinal borders of the relevant straight DB section which experimentally are less precisely identified.

## Data availability
The datasets generated during and/or analysed during the current study are available from the corresponding author on reasonable request.

## Code availability
The code used in the current study is available from the corresponding author on reasonable request.

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

## Acknowledgements
The authors are indebted to Werner Hanke for enlightening discussions and to Johannes Weis for helpful technical assistance. This work was supported by the Deutsche Forschungsgemeinschaft (DFG) through the Würzburg-Dresden Cluster of Excellence on Complexity and Topology in Quantum Matter—ct.qmat (EXC 2147, project-id 390858490) and the Collaborative Research Center SFB 1170 "ToCoTronics" (project-id 258499086). The authors gratefully acknowledge further support by the National Key R&D Program of China (2017YFE0131300) and by the Sino-German mobility program (M-0006). This publication was supported by the Open Access Publication Fund of the University of Wuerzburg.

## Author contributions
R.S., A.K., and F.R. conducted the epitaxial growth and surface characterization and carried out the STM experiments. R.S. and A.K. analyzed the STM/STS data. D.J., F.D., B.S. performed the theoretical tight-binding calculations on the basis of DFT calculations conducted by G.L. . E.M.H. supervised the efforts of theoretical tight-binding calculations. J.S. and R.C. supervised the experiments. R.C. conceived and led the project. R.S. and R.C. wrote the manuscript with input from all other authors.

## Funding

## Competing interests
The authors declare no competing interests.
