## [Peer Review File · Nature Communications]

Effective lifting of the topological protection of quantum spin Hall edge states by edge couplingREVIEWER COMMENTS

Reviewer #1 (Remarks to the Author):

This manuscript reports the observation of the quasiparticle interference signals in the topological edge modes of the 2D topological insulator bismuthene. In the case of the isolated edge, the quasiparticle interference should not occur because the spin-momentum locking prevents the backscattering. The authors focus on the domain boundary, which acts as a line defect. Because the counterpropagating state with the same spin is available across the domain boundary, the backscattering may be allowed, and a segment of the domain boundary can act as a 1D Fabry-Prot resonator. By using STM/STS, the authors indeed observed the quasiparticle-interference-like signals in the conductance images.

Control of the topological edge state is an important topic, and the idea of coupling the edge states across the domain boundary is interesting. Therefore, I assess that the manuscript is potentially appropriate for Nature Communications. However, I think that the authors should improve the data quality and perform additional experiments/analyses to corroborate the quasiparticle interference scenario. I recommend that the authors consider the following comments.

1. The observed interference patterns are not so clear to me. In Figs. 3f and 3g, modulations at E4 and E3 are apparent but others are not. I agree that the data are somehow consistent with the quasiparticle interference and understand that it is challenging to acquire higher quality data. Nevertheless, more convincing data are necessary to justify the publication in Nature Communications.
2. In Fig. 3f, the spectrum at the kink is similar to the one in the straight segment where the interference-like signals are observed. This may mean that the kink allows the electrons to go through. Please elaborate on how the kinks form the resonator even though they affect little on the electronic state.
3. The authors plot the data points for 8 different domain boundaries in Fig 4a. I recommend that the authors show the dI/dV maps of all the domain boundaries in the Supplementary Information.
4. The segment-length dependence of the Fabry-Prot resonance should be more systematically investigated and analyzed. For example, the energy separation between the Fabry-Prot modes may become smaller as the length of the segment gets longer. Is this observed?
5. Minor technical comment: the authors write in the Methods section that there is a Gaussian energy broadening associated with the external modulation. The broadening function due to the lock-in modulation has a semicircular shape rather than a Gaussian shape.

Reviewer #2 (Remarks to the Author):

see attachment

Reviewer #3 (Remarks to the Author):

The authors report STM/STS studies of coupled edge states at antiphase domain boundaries (DBs) of bismuthene, a single layer honeycomb lattice of bismuth synthesized on SiC substrate. Following their prior work on bismuthene where a large band gap and edge state were observed, the authors observed standing-wave-like quasi-particle interference (QPI) pattern at discrete energies at short antiphase DBs with kinks as scattering centers. The observed QPI was interpreted as the scattering between topological edge states from different antiphase domains, providing evidence of the topological nature of the edge states and the bulk states (2D TI). The results are reasonably convincing given the challenges of synthesizing high quality 2D TI and fewer experimental studies of coupling between topological edge states. There are a few lingering issues before I could recommend publication of the manuscript.

1. The authors listed 9 DBs in Fig. S4. However, they only show one spectroscopy data set (Fig. 3). It would be more convincing to show additional spectroscopy data sets of other DBs in supplementary information.

2. From the data, it seems all DBs have the same channel width. Did the authors find any DB with different (presumably larger) width? If so, would that DB show a weaker coupling of edge state because of less overlap? This would give some information on the half width of the topological edge states.

3. What causes the variation of local chemical potential shown in Fig. S4? Is it from the substrate or from defects/impurities inside bismuthene?

4. For free zigzag edge, the LDOS of edge states shows Luttinger liquid behavior, which is a collective electronic states due to strong correlations. But interpretation of QPI is the standing wave of coupled edge states with linear dispersion, which is a single particle picture without correlation. How to reconcile these two pictures? Does the coupling of edge states "destroy" the LL state so that the single particle states are recovered?

5. As shown in Fig. S7, the coupling of edge states open a hybridization gap (0.12 eV), which presumably should affect the lowest quantum well (standing wave) state in long DBs (e.g. the one shown in Fig.S3) such that the dispersion deviates linear relationship. However, the data points in Fig. 4a are all from relatively short DBs. It would be more interesting and more convincing to show indication of hybridization gap from QPI of long DBs.

6. A minor comment, the tight binding modeling in supplementary uses a stripe of trivial insulator inside TI ribbon, effectively a "particle in a box picture" for coupled edge states. There is no antiphase between two sides of the "line defect", which is different from the experimental situation. Is the antiphase relationship important for the QPI? Can the authors build a tight binding model of antiphase DB?

Reviewer #1

This manuscript reports the observation of the quasiparticle interference signals in the topological edge modes of the 2D topological insulator bismuthene. In the case of the isolated edge, the quasiparticle interference should not occur because the spin-momentum locking prevents the backscattering. The authors focus on the domain boundary, which acts as a line defect. Because the counterpropagating state with the same spin is available across the domain boundary, the backscattering may be allowed, and a segment of the domain boundary can act as a 1D Fabry-Pérot resonator. By using STM/STS, the authors indeed observed the quasiparticle-interference-like signals in the conductance images.

Control of the topological edge state is an important topic, and the idea of coupling the edge states across the domain boundary is interesting. Therefore, I assess that the manuscript is potentially appropriate for Nature Communications.

We thank the reviewer for the appreciation of the key idea of our work and for acknowledging that the control of the topological edge states in QSH insulators is an important topic.

However, I think that the authors should improve the data quality and perform additional experiments/analyses to corroborate the quasiparticle interference scenario. I recommend that the authors consider the following comments.

1. The observed interference patterns are not so clear to me. In Figs. 3f and 3g, modulations at E4 and E3 are apparent but others are not. I agree that the data are somehow consistent with the quasiparticle interference and understand that it is challenging to acquire higher quality data. Nevertheless, more convincing data are necessary to justify the publication in Nature Communications.

We agree with the reviewer that the LDOS maps in Figs. 3f and 3g are challenging to read due to their packed spatial and spectroscopic information. This partly results from the fact that the interference pattern of the (Bloch-like) edge states is additionally modulated by the atomic corrugation. The clearest identification of the interference modulations can be seen from the dI/dV constant energy maps in Fig. 3g and, in particular, the associated dI/dV line profiles, integrated over the lateral extent of the domain boundary (y-direction) and plotted on top of the respective constant energy maps. In order to better highlight the standing wave behavior of the levels E1-E4 we have added grey-shaded regions to Fig. 3g as guide to the eye: the space between them marks the effective resonator. Furthermore, in order to demonstrate the ubiquitous and fully reproducible character of these observations we have added the corresponding data sets for nine more representative domain boundary segments to the Supplemental Information (**Figs. S9a – h and Fig. S11**). We are confident that these data and their systematic behavior can convincingly demonstrate the existence of the Fabry-Pérot modulations inside the domain boundaries.

2. In Fig. 3f, the spectrum at the kink is similar to the one in the straight segment where the interference-like signals are observed. This may mean that the kink allows the electrons to go through. Please elaborate on how the kinks form the resonator even though they affect little on the electronic state.

We are not quite sure, which spectrum and which kink the referee is specifically referring to. Nonetheless, she/he is absolutely correct that the edge state electrons are not perfectly reflected by the kinks and that there is a finite transmission probability, just as in an optical Fabry-Pérot interferometer. The kinks (defined by the locations where a domain boundary changes direction) as well as other atomic defects (we show one example in Fig. S11 in the Supplemental Information) are local, i.e., 0D defects which can be thought of as point-like scatterers for the 1D (coupled) edge states. The effect can be captured by a complex reflectivity $R = |R|e^{i \arg(R)}$, whose amplitude will generally be less than unity ($|R| < 1$), with both $|R|$ and the phase factor $\arg(R)$ depending on the microscopic details of the respective scattering potential. The resulting dI/dV spectra are thus composed of two components: (1) a rather smooth background signal associated with the unscattered, i.e., transmitted part of the propagating edge state, and (2) the strongly modulated (along the domain boundary, Figs. 3f and g) and peaky (in energy, Fig. 3d) component resulting from the backscattering-induced interference. Hence, it does not come as a surprise that the dI/dV signal at the kinks still displays some finite LDOS. They nonetheless represent nodal points for the interference modulations (see Fig. 3g, and Fig. S11c, e).

The existence of an unscattered (i.e., transmitted) partial wave is also the reason, why we refer to the observed interference patterns as "Fabry-Pérot resonances", rather than as "quantum well states" (which are typically defined by perfectly reflecting potential walls). This circumstance can be well understood by the measurement shown in Fig. S11, a longer domain boundary that has a defect inside, and which is strongly reminiscent of a coupled double Fabry-Pérot resonator. In this context, and directly connected to the referee's question, we have built a simple model for this system. In this model (discussed around Fig. S11 in the Supplemental Information) we demonstrate that an effective resonator model with scattering centers characterized by sub-unity reflectivity and hence finite transmission probability reproduces good qualitative agreement with our observations. We note however, that an ab-initio atomistic modelling of the kinks and the associated potential landscape is not addressed here.

3. The authors plot the data points for 8 different domain boundaries in Fig 4a. I recommend that the authors show the dI/dV maps of all the domain boundaries in the Supplementary Information.

We thank the reviewer for this valuable suggestion. As already mentioned in our response to point #1., we now have added the data sets of all these domain boundaries to the Supplemental Information (Figs. S9a-h). Additionally, we provide data from a long domain boundary that exhibits an atomic point defect within, and also demonstrates strong Fabry-Pérot oscillations (Fig. S11). We think that taking all the collected and presented data into account, we convincingly demonstrate the existence of Fabry-Pérot modulations along the domain boundaries, caused by kinks or atomic point defects acting as scattering centers.

4. The segment-length dependence of the Fabry-Pérot resonance should be more systematically investigated and analyzed. For example, the energy separation between the Fabry-Perot modes may become smaller as the length of the segment gets longer. Is this observed?

This is indeed a very interesting point, and we are grateful to the referee for encouraging to us to include such an analysis. In the new Fig. S10 of the Supplementary Information we evaluate the mean energy spacing DE of the Fabry-Pérot resonances as a function of the domain boundary length L . Indeed, it shows a systematic decrease of the level spacing roughly following the expected $\propto 1/L$ dependence. We should note, however, that there lies some ambiguity in the

determination of the resonator length, as the scattering centers (kinks or atomic defects) have some finite extension. Modeling the associated scattering potential by a point-like scatterer (and the relevant complex reflectivity) will then render an *effective* resonator length L whose quantitative relation to the actual atomic structure would require an ab-initio calculation of the potential landscape. The L values used in the above analysis are estimated from the STM topographies (i.e., from the detected atomic structure), and hence have to be taken with some caution.

This uncertainty is also the reason why in our evaluation of the edge state dispersion relation (presented in Fig. 4a of the paper) we refrain from using the domain boundary lengths L as reference points. Instead, we employ an unbiased method and directly infer the spatial separation of modulation maxima λ as indicated in the dI/dV line profile of level E4 in Fig. 3g, thus avoiding any a priori assumptions on L .

5. Minor technical comment: the authors write in the Methods section that there is a Gaussian energy broadening associated with the external modulation. The broadening function due to the lock-in modulation has a semicircular shape rather than a Gaussian shape.

We thank the referee for this comment and corrected this remark in the methods section. Our prior formulation was based on a common approximation for the resulting spectral broadening, i.e., the common approximation of a convolution with a semicircle with radius V_m , by a convolution with a gaussian function with FWHM of $2.5 \cdot V_m$. We note, however, that this has no implications for our analysis, since we do not rely on directly fitting dI/dV spectra as a function of energy.

Reviewer #2

R. Stühler and co-authors report on low-temperature scanning tunneling microscopy (STM) and spectroscopy (STS) measurements on the edges and surface of the 2D TI bismuthene. Previously, the authors had reported the synthesis and STM study of bismuthene, which exhibits a one-dimensional topological edge state along its boundaries. The current study under review addresses the electronic properties of various edge geometries and domain boundaries, which are formed between two nearby edges. Measuring quasiparticle interference patterns along such a domain boundary, the authors observe a standing wave pattern in the electronic density of states of the edge state. The appearance of this pattern is interpreted as a signature of quasiparticle backscattering, which is induced by the coupling between the two edge states.

The well-written manuscript touches on an interesting topic and the experimental data are of high quality.

We appreciate that the Referee finds our manuscript well written and that he/she acknowledges the high quality of our data.

Nevertheless, it suffers from a lack of novelty and an incorrect interpretation of the experimental data, which prevents me from recommending its publication.

Obviously, we cannot follow this assessment. Rather, we strongly believe that it is primarily based on an unfortunate misconception on the Referee's side, namely an erroneous picture of the band dispersion at the bismuthene zigzag edges. Below we lay down explicitly what misunderstanding of our system has unfortunately led the Referee to her/his incorrect conclusions.

Generally, topological protection is provided by the bulk symmetries, such as time-reversal symmetry (TRS) in the present case. Experimentally, such protection can manifest in quantized electronic transport along a one-dimensional topological edge state and the absence of quasiparticle backscattering in STM experiments. In case TRS is broken, e.g., by the application of an external magnetic field, the topological edge state can couple, which results in a spectral gap. If Fermi energy is placed inside this gap, electronic transport along the edge state is suppressed. In a local picture, the placement of magnetic atoms or nanostructures, which locally break TRS, on a topological edge state can induce backscattering. This process can be best understood in terms of spin-flip scattering at the magnetic moment that converts a forward moving spin-up in backward moving spin-down quasiparticles. In fact, this phenomenon, quasiparticles backscattering induced by magnetic clusters adsorbed to the topological edge state of bismuth has already been observed (PNAS 117, 16214-16218 (2020)).

We fully agree with the reviewer that TRS breaking (by, e.g., an external magnetic field or by magnetic impurities) will allow for spin-flip scattering and, consequently, for quasiparticle backscattering. In contrast, however, in our case TRS remains perfectly preserved, and yet (spin-preserving) backscattering can be induced by spatial coupling of topological edges states. It is precisely this feature of our observations, which defines their novelty and their potential for possible device applications. By comparison, controlling topological protection of QSHI edge states (or rather their backscattering) by magnetic means seems far less feasible for actual device applications. At least, we know of no device concepts along this line – in contrast to those based on TRS-preserving inter-edge coupling, as cited in the opening paragraphs of our paper.

Considering the manuscript under review, the coupling of a topological edge state to any other electronic state, topological or not, will in most cases open additional scattering channels. The presence of additional (spin-degenerate) bands will allow quasiparticles to (back)scatter from the 1D topological edge state to the other electronic states. Hence, topological protection is not lifted in the actual sense. We are rather looking at the consequences of a trivial hybridization effect on quasiparticles scattering. This effect can occur between any type of band structure.

There is nothing in these statements that we cannot agree on. The hybridization (or tunneling) between opposite edges of a QSH insulator may indeed be perceived as a trivial effect.

What is more concerning is the analysis and interpretation of the experimental data in Fig. 3f-h and Fig. 4. The authors argue that the standing wave patterns observed in Fig. 3g present evidence for backscattering between two topological edge states each of which is characterized by a Dirac cone band structure. However, this interpretation is incorrect. Consider the calculated band structure of two coupled bismuthene zig zag edges shown in Fig. S7. It shows a spectral gap at the former Dirac point as well as linearly dispersing almost spin-degenerate bands to higher and lower energies. For such a band structure, one expects a quasiparticle dispersion and associated real space local density of states amplitude as shown in the attached figure Ref.1 below. More specifically, one would observe a QPI pattern that is symmetric about the former Dirac point, i.e., QPI in an electron-like and hole-like band pocket.

Clearly, that is different from the author's observations. While the authors do observe a spectral gap (Fig. 3d) that could indicate the existence of a gapped-out Dirac point, the observed QPI pattern does not fit that picture. Instead, the authors observe a monotonically increasing QPI dispersion (Fig. 3h and 4a) starting from $E \sim -0.3$ eV below the putative Dirac point to energies just above it at $E \sim E_F$. This monotonic increase of the scattering vector amplitude with energy (Fig. 3h and Fig. 4a) directly excludes the existence of a gapped Dirac point in the band structure of the domain boundary. Instead, the dispersion (Fig. 3h) and real space QPI pattern (Fig. 3g) rather indicate the existence of a single electron-like band pocket around E_1 (-0.3 eV). It follows that the domain boundary does not host a pair of Dirac-like topological edge states in the bulk gap and the author's claim to lift topological protection by edge-coupling is incorrect.

We thank the Referee for illustrating his above arguments by a figure ("Fig.Rev 1", reproduced below) which clearly indicates that her/his criticism is based on a misconception of the underlying band dispersion: the zigzag edge states do NOT disperse symmetrically about the Dirac point! This is clearly demonstrated by the ab-initio DFT ribbon calculations in Ref. [L1], see Fig. L1-S5 below. Clearly, the dispersion of the zigzag edges is highly asymmetric, with an extended linear dispersion above the Dirac point, but rather flat behavior below E_D . The latter effect will squeeze any energy quantization to a very narrow band width and make it difficult to identify individual Fabry-Pérot resonance peaks. Furthermore, because the Dirac point is located below the valence band maximum, this energy range will be dominated by the edge-projected bulk band structure. Hence, the Fabry-Pérot states consistently observed in all our measurements are derived from the quasi-linearly dispersing branch **ABOVE** the Dirac point. We have added a sentence to the paper which clarifies the highly asymmetric band situation.

The referee may have been misguided by the tight-binding (TB) band structure in Fig. S7 of the Supplementary Information, which at first glance might suggest symmetric behavior, at least locally about the Dirac point. However, we wish to point out that the TB model only serves to provide a qualitative picture for edge coupling, but cannot reliably account for the precise band dispersion.

Fig. Rev 1
 sketch from report of
 Referee#2

Fig. L1-S5
 DFT calculation
 (from Ref. [L1])

Further, we wish to comment that the energy scale which the Referee refers to is incorrect: she/he states that "the monotonically increasing QPI dispersion (Fig. 3h and 4a) starts from $E \approx -0.3$ eV **below the putative Dirac point**". We have difficulties following this statement: the starting point of the linear dispersion is precisely the Dirac point, i.e., these two figures show that the Dirac point is located at almost 0.3 eV below the VBM (Fig. 4a) and slightly more than 0.4 eV below E_F (Fig. 3h), fully consistent the observation of our Fabry-Pérot resonances (E_1, E_2, \dots) lying above these energies! The position of the Dirac point is therefore NOT "putative", but experimentally determined.

Other comments

1. The authors have omitted a whole body of relevant references on quasiparticle (back-)scattering in topological edge states observed with the STM. The authors should update their references accordingly.

The referee pointed out the recent paper of backscattering induced by magnetic impurities (PNAS 117, 16214-16218 (2020)), which is relevant to our study only on a very general scope, as it relies on the need to break TRS. We therefore cite it now in the introductory part of the main text. We wish to reiterate, however, that we did not intend to study the effect of backscattering induced by TRS breaking, **but rather and specifically in the absence of it!** For this situation we are certain that we have cited all relevant references.

2. The authors should use absolute units for the apparent height (z) and local density of states (dI/dV) measured with the STM.

The information of apparent height (z) in absolute units is of no relevance to our findings. We have omitted it throughout the main paper in favor of better readability, especially for not having to include a different color bar for every single constant current image.

The information of the dI/dV in absolute units is also of no relevance to our findings. We have measured STS with a standard lock-in technique. Therefore, all spectra do not exhibit an absolute scale for dI/dV in units of nS .

Reviewer #3:

The authors report STM/STS studies of coupled edge states at antiphase domain boundaries (DBs) of bismuthene, a single layer honeycomb lattice of bismuth synthesized on SiC substrate. Following their prior work on bismuthene where a large band gap and edge state were observed, the authors observed standing-wave-like quasi-particle interference (QPI) pattern at discrete energies at short antiphase DBs with kinks as scattering centers. The observed QPI was interpreted as the scattering between topological edge states from different antiphase domains, providing evidence of the topological nature of the edge states and the bulk states (2D TI). The results are reasonably convincing given the challenges of synthesizing high quality 2D TI and fewer experimental studies of coupling between topological edge states. There are a few lingering issues before I could recommend publication of the manuscript.

We thank the Referee for her/his appreciation of our work and the key ideas we wish to convey. We are confident that we can clarify all Referee questions in a satisfactory manner.

1. The authors listed 9 DBs in Fig. S4. However, they only show one spectroscopy data set (Fig. 3). It would be more convincing to show additional spectroscopy data sets of other DBs in supplementary information.

We thank the Referee for this suggestion and followed her/his advice by showing the corresponding data sets in Figs. S9a – h in the Supplementary Information. Additionally, in Fig. S11 we provide data from a long domain boundary that exhibits a point defect within. We think that taking all the collected and presented data into account, we convincingly demonstrate the existence of the Fabry-Pérot modulations inside the domain boundary, and that defects, such as kinks or point defects act as scattering centers (with finite reflectivity).

2. From the data, it seems all DBs have the same channel width. Did the authors find any DB with different (presumably larger) width? If so, would that DB show a weaker coupling of edge state because of less overlap? This would give some information on the half width of the topological edge states.

This is a very nice and relevant question, which we have been wondering about ourselves. If the width of the domain boundary would be increased, we would indeed expect a weaker hybridization of the edge states. However, we have consistently observed only one kind of domain boundary, which always exhibits the very same width as a result of the epitaxial growth.

As for the spatial (lateral) extent of the topological edge state: this has already been determined in our previous STM study [L1]. It has been determined to be of exponential decay from the edge into the 2D bismuthene bulk, with a $1/e$ decay constant of ≈ 4 Angstroms. This is consistent with the known values of bandgap and Fermi velocity, and also compatible with the hybridization effects seen in our domain boundaries.

3. What causes the variation of local chemical potential shown in Fig. S4? Is it from the substrate or from defects/impurities inside bismuthene?

The variation of local chemical potential is attributed to both the n-doping of the SiC substrate and the local defect/impurity concentration in the bismuthene film. Firstly, the nitrogen doping centers in the substrate display some density variations on length scales of $\approx 10 - 100$ nm, i.e., on a

relevant length scale of our investigations. We have recently observed similar effects also on other SiC-supported monolayers [L4]. Secondly, the bismuthene film exhibits local variation of film inhomogeneities, stemming from defects or impurities, which can be seen from Fig. 2a. It is to be expected that these variations also may cause small local fluctuations of the chemical potential.

4. For free zigzag edge, the LDOS of edge states shows Luttinger liquid behavior, which is a collective electronic states due to strong correlations. But interpretation of QPI is the standing wave of coupled edge states with linear dispersion, which is a single particle picture without correlation. How to reconcile these two pictures? Does the coupling of edge states “destroy” the LL state so that the single particle states are recovered?

The referee points towards an interesting aspect of many-particle physics observed in bismuthene, see Ref. [L2]. Our understanding is that, first, the Luttinger liquid physics is only relevant in a low-energy sector (around the Fermi energy), i.e. ± 100 meV, as seen in Ref. [L2]. In the case of the observed Fabry-Pérot interferences, which involve only single-particle scattering processes, the energy scales are several hundreds of meV. Thus, any Luttinger liquid physics would not disturb most of the observed interference levels.

Secondly, we note that a domain boundary is a finite structure, where the Luttinger liquid physics may break down and other physical finite size related phenomena, such as Coulomb blockade, might dominate or obscure the signatures of Luttinger liquid physics. These are nonetheless interesting question being subject to ongoing and future studies.

5. As shown in Fig. S7, the coupling of edge states open a hybridization gap (0.12 eV), which presumably should affect the lowest quantum well (standing wave) state in long DBs (e.g. the one shown in Fig.S3) such that the dispersion deviates linear relationship. However, the data points in Fig. 4a are all from relatively short DBs. It would be more interesting and more convincing to show indication of hybridization gap from QPI of long DBs.

First, we want to point out that, since the tight-binding calculation only models the domain boundary qualitatively, the value for the hybridization gap (0.12 eV) from the tight-binding modeling might not be quantitatively precise. It is likely that the real value, which could be substantially smaller than 0.12 eV, is strongly dependent on the exact atomic configuration within the domain boundary, which is not explicitly taken into account in the tight-binding model. Conceptually, however, we point out that the formation of a hybridization gap must be present for QPI to be observed in this system hosting helical edge states.

For a better identification of the hybridization gap it would be desirable to have very long defect-free domain boundaries, where the spectra are not dominated by the energetically discretized Fabry-Pérot states. Unfortunately, though, these situations are extremely rare. If longer domain boundaries are observed at all, they typically exhibit additional point defects inside, as shown in Figs. S3 and S11.

For the latter case we were able to perform a spectroscopic STS measurement. It turns out that the point defect inside the domain boundary already induces backscattering, similar to what is observed at kinks. Therefore, we think it is very unlikely that on an appropriate time scale we will be able to measure a significantly longer AND defect free domain boundary.

6. A minor comment, the tight binding modeling in supplementary uses a stripe of trivial insulator inside TI ribbon, effectively a “particle in a box picture” for coupled edge states. There

is no antiphase between two sides of the "line defect", which is different from the experimental situation. Is the antiphase relationship important for the QPI? Can the authors build a tight binding model of antiphase DB?

We thank the referee for this interesting question. She/he is, of course, fully correct that the TB model differs with respect to the atomistic experimental scenario. Our TB model only models the atomic lattice of the bismuthene layer but not that of the underlying substrate. By its very construction it thus does not contain any phase relationships (or "antiphase relationship" as the referee puts it) between Bi layer and substrate. The main purpose of this model was to provide a simple picture of what happens qualitatively when two bismuthene zigzag edges in close proximity start to couple with each other.

The atomistic Hamiltonian from ab-initio calculations of the DBs may be unknown, but we can still use our simple TB model to mimic its trivial insulating phase by tuning the hopping parameters. Any structural details of the DB, like the antiphase relationship, can be mimicked by simply adjusting these parameters. The strength of the coupling between opposite edges (i.e., the electronic tunneling between them) will generally be governed by the overlap of the exponential tails of the edge states on either side. Due to covalent coupling of the bismuthene film to the SiC substrate ("orbital filtering") we may expect that the lateral phase relationship between neighboring domains could play an additional role for a QUANTITATIVE understanding of the coupling strength. This very interesting aspect can be addressed in future ab-initio studies of this effect, but following the above reasoning, we expect to observe the same QUALITATIVE behavior of the Fabry-Pérot states, independently of the structural details of the system.

FIG. S9. **a – c, Fabry-Pérot states in different domain boundaries with varying lengths.** dI/dV resonance data for eight domain boundary segments (see continuation of Fig. S9 in Figs. S9d – f, and S9g – h). **a0 – h0**, Constant current STM images of the domain boundaries. The length L of each domain boundary is indicated inside the respective image. The L values are estimated from the STM topographies (i.e., from the detected atomic structure) and are subject to an estimated uncertainty of $\approx \pm 0.5\text{nm}$. **ai – hi**, $i \geq 1$, The differential conductivity maps plotted above each constant current image are taken within the region of the respective domain boundary marked by the red rectangle in Figs. a0 – h0, and correspond to the i^{th} Fabry-Pérot resonance level. The respective energy with respect to E_F is indicated on the left side of each panel. The differential conductivity line profiles resulting from integrating over the width of the domain boundary in the corresponding differential conductivity map are shown above each map. The arrows indicate the positions of resonance maxima, which are obtained by fitting the data in the vicinity of each resonance maximum with a Gaussian function (red dashed lines indicate best fits). Scan parameters are listed in Tab. S1.

FIG. S9. d – f, Fabry-Pérot states in different domain boundaries with varying lengths; continuation.

FIG. S9. g – h, Fabry-Pérot states in different domain boundaries with varying lengths; continuation.

FIG. S10. Mean Fabry-Pérot resonance spacing DE as a function of DB length L . The error bars indicate the std in the case of DBs, where more than two Fabry-Pérot resonances are observed. The red line indicates a a/L function with $a = 0.965 \text{ eV nm}^{-1}$.

FIG. S11. **a**, Constant-current STM measurement of a DB with a defect inside, that splits the DB into two straight segments with different lengths. The red rectangle indicates the region for spatially resolved dI/dV mapping. Scanning parameters: $V_{\text{set}} = -1.2$ V, $I_{\text{set}} = 10$ pA, $T = 4.35$ K. **b**, dI/dV integrating over the width of the DB as a function of longitudinal position x . Energy dependent charge modulations are observed both in the left and right DB segment, for which towards higher energies their wavelengths shortens. The situation is reminiscent of a coupled double Fabry-Pérot resonator, see **d**, **e** for qualitative modelling. **c**, dI/dV maps at constant energy marked with arrows in **b**. For certain energies the charge modulations are mainly confined within the left segment of the DB and for certain other energies mainly confined within the right segment of the DB. The scale bar is 2 nm. **d**, Schematic of coupled double Fabry-Pérot resonator for qualitative modelling of the observed charge modulations. The coupled resonator is composed of a first resonator with length L_1 and a second resonator with length L_2 . Both resonators are coupled with a partially reflective wall R_0 . Charge carriers injected into the resonator by the STM tip are reflected/transmitted multiple times at the resonators walls that are characterized by a reflectivity $R_i = |R_i|e^{i\arg(R_i)}$, where $i = 0, 1, 3$. In a plane wave scattering approach the electronic wave function ψ at position x can be modelled by summing over all possible scattering paths, see Eqs. (S12)–(S19) (the initial amplitude of the injected carrier is set to 1). **e**, Simulation of charge modulations in a coupled double Fabry-Pérot resonator in **d**. The false color plot shows $|\psi|^2$ from Eq. (S12) for model parameters: $L_1 = 10.0$ nm, $L_2 = 57.0$ nm, $R_0 = 0.7e^{i\pi}$, $R_1 = R_2 = 0.5e^{i\pi}$, and $E_{R/L} - E_F = \hbar v_F k_{R/L} + \mu$, where $\hbar v_F = 3.6$ eV Å and $\mu = -0.56$ eV. The infinite sums in Eqs. (S13), (S14) were evaluated up to $j_F = 5$ and $m_F = 2$.

[L1]: Reis, F. et al. Bismuthene on a SiC substrate: “A candidate for a high-temperature quantum spin Hall material”. *Science* **357**, 287 - 290 (2017). <https://doi.org/10.1126/science.aai8142>

[L2]: Stühler, R. et al. „*Tomonaga-Luttinger liquid in the edge channels of a quantum spin Hall insulator*”. *Nature Physics* **16**, 47 - 51 (2020). <https://doi.org/10.1038/s41567-019-0697-z>

[L3]: J. Seo, P. Roushan, H. Beidenkopf, Y. S. Hor, R. J. Cava, and A. Yazdani, “*Transmission of Topological Surface States Through Surface Barriers*”. *Nature* **466**, 343 (2010). <https://doi.org/10.1038/s42005-021-00657-6>

[L4]: Bauernfeind, M., Erhardt, J., Eck, P. et al. “*Design and realization of topological Dirac fermions on a triangular lattice*”. *Nat Commun* **12**, 5396 (2021). <https://doi.org/10.1038/s41467-021-25627-y>

REVIEWER COMMENTS

Reviewer #1 (Remarks to the Author):

The authors addressed the issues pointed out by the referees and adequately revised the manuscript. There is a typo in the amended part (Methods section, $V_{\text{mod}}^2+V^2$ should be $V_{\text{mod}}^2-V^2$). I do not find other issues and am happy to recommend the publication of the revised manuscript in Nature Communications.

Reviewer #2 (Remarks to the Author):

I appreciate the author's detailed response to my comments. They argue that my criticism of their interpretation is based on my misconception of the underlying band dispersion. Indeed, I share the author's argument that model calculations can in many cases only provide a qualitative picture of the electronic structure. This may be particularly true for the rather complicated structure of a domain boundary on a substrate, such as discussed in the manuscript under review. As can be seen from my previous comments, my arguments are solely based on the experimental data presented by the authors in Fig. 3 and 4 and they are not misguided by the tight-binding model. I'll take this opportunity to reiterate this reasoning.

The local density of states (LDOS) of the free zig-zag edge shown in Fig. 3a vanishes near zero energy inside the bulk band gap and increases (more or less) monotonically to positive and negative energies. If a helical edge state Dirac point, which is characterized by a vanishing LDOS, is situated in the energy window shown in Fig. 3a, it must, therefore, reside near zero voltage. I think this is an obvious argument which the authors will share after calculating the LDOS of a Dirac cone. Hence, the author's statement that the position of the Dirac point was 'experimentally determined' to reside at 'slightly more than 0.4 eV below EF' is simply not correct, and I wonder which experimental data presented in the paper lead to their conclusion.

Nevertheless, the authors correctly write that the coupling between two helical edge modes can gap out the Dirac point, e.g., as sketched in the cartoon of Fig. 1b. At first, the experimental data on the domain boundary LDOS presented in Fig. 3b seem to be consistent with that expectation in that the area around zero energy has developed a gap. If a gapped Dirac point existed around zero energy, one would expect a QPI pattern similar to the one drawn in my previous report, that is, an (electron) hole like dispersion (above) below the gapped out Dirac point. It is important to note that even in case the dispersion below the Dirac point were non-linear and rendered the QPI observation challenging, as the authors argue, one would still expect to observe the electron-pocket like dispersion above the Dirac point at zero energy. These expectations are in obvious contradiction to the linearly dispersing QPI crossing zero energy reported by the authors.

Hence, I can uphold my previous assessment: The presented experimental LDOS and QPI data are inconsistent with the existence of a gapped Dirac cone in the domain boundary in the presented energy window. It follows that the conclusion by the authors on induced back-scattering is not supported by experimental data and I recommend revising their interpretation based on their experimental data rather than ab-initio calculations. The electronic structure of the domain boundary remains an interesting question to be addressed in more detail in the future. Without touching on the questions of novelty or impact, it is on the basis of these arguments that I cannot recommend publication of this work in Nature Communications.

Other major comment

I appreciate that the authors share my opinion. It is the breaking of bulk symmetries that lifts topological protection. While the coupling to other modes—non-trivial or not—can induce backscattering, it does not break or lift the topological protection of a quantum spin Hall insulator (=the bulk), as the authors claim in the title. Therefore, the manuscript's title and some of its content are misleading, if not physically incorrect, and should be amended (.backscattering induced by edge-hybridization' or so would be more appropriate...).

Minor comment

While it is my understanding that the presentation of experimental data in their actual units is preferable, I do respect the author's choice. Nevertheless, I insist on the labeling of the STM topography's apparent height at one position in the manuscript, because this could be a relevant piece of information for readers who are not yet familiar with the investigated material platform. Fig. 2a or d seem to be suitable for that purpose.

Reviewer #3 (Remarks to the Author):

The authors have addressed my comments satisfactorily in their replies and the revised manuscript. I also think they properly address the criticisms from other reviewers. The additional data in supplementary materials support the results and conclusion in the main text. I believe this manuscript is ready for publication in Nature communications.

Reviewer #1 and Reviewer #3 (Remarks to the Author):

The authors addressed the issues pointed out by the referees and adequately revised the manuscript. There is a typo in the amended part (Methods section, $V_{\text{mod}}^2+V^2$ should be $V_{\text{mod}}^2-V^2$). I do not find other issues and am happy to recommend the publication of the revised manuscript in Nature Communications.

The authors have addressed my comments satisfactorily in their replies and the revised manuscript. I also think they properly address the criticisms from other reviewers. The additional data in supplementary materials support the results and conclusion in the main text. I believe this manuscript is ready for publication in Nature communications.

We thank both reviewers for their positive response and their recommendation for acceptance. We corrected the mentioned typo in the Methods section.

Reviewer #2 (Remarks to the Author):

I appreciate the authors' detailed response to my comments. They argue that my criticism of their interpretation is based on my misconception of the underlying band dispersion. Indeed, I share the authors' argument that model calculations can in many cases only provide a qualitative picture of the electronic structure. This may be particularly true for the rather complicated structure of a domain boundary on a substrate, such as discussed in the manuscript under review. As can be seen from my previous comments, my arguments are solely based on the experimental data presented by the authors in Fig. 3 and 4 and they are not misguided by the tight-binding model. I'll take this opportunity to reiterate this reasoning.

The local density of states (LDOS) of the free zig-zag edge shown in Fig. 3a vanishes near zero energy inside the bulk band gap and increases (more or less) monotonically to positive and negative energies. If a helical edge state Dirac point, which is characterized by a vanishing LDOS, is situated in the energy window shown in Fig. 3a, it must, therefore, reside near zero voltage. I think this is an obvious argument which the authors will share after calculating the LDOS of a Dirac cone. Hence, the authors' statement that the position of the Dirac point was "experimentally determined" to reside at "slightly more than 0.4 eV below E_F " is simply not correct, and I wonder which experimental data presented in the paper lead to their conclusion.

Nevertheless, the authors correctly write that the coupling between two helical edge modes can gap out the Dirac point, e.g., as sketched in the cartoon of Fig. 1b. At first, the experimental data on the domain boundary LDOS presented in Fig. 3b seem to be consistent with that expectation in that the area around zero energy has developed a gap. If a gapped Dirac point existed around zero energy, one would expect a QPI pattern similar to the one drawn in my previous report, that is, an (electron) hole like dispersion (above) below the gapped out Dirac point. It is important to note that even in case the dispersion below the Dirac point were non-linear and rendered the QPI observation challenging, as the authors argue, one would still expect to observe the electron-pocket like dispersion above the Dirac point at zero energy. These expectations are in obvious contradiction to the linearly dispersing QPI crossing zero energy reported by the authors.

Hence, I can uphold my previous assessment: The presented experimental LDOS and QPI data are inconsistent with the existence of a gapped Dirac cone in the domain boundary in the presented energy window. It follows that the conclusion by the authors on induced back-scattering is not supported by experimental data and I recommend revising their interpretation based on their experimental data rather than ab-initio calculations. The electronic structure of the domain boundary remains an interesting question to be addressed in more detail in the future. Without touching on the questions of novelty or impact, it is on the basis of these arguments that I cannot recommend publication of this work in Nature Communications.

The central disagreement between Reviewer #2 and us concerns the energy location of the Dirac point (DP) of the topological edge state: the referee seems firmly convinced that it is positioned at zero bias voltage. This is simply an incorrect and unfounded assertion, as can be seen by the following arguments:

- (1) There exists no general symmetry argument or other physical reason why the DP of a topological boundary state should be placed exactly at the Fermi level.
- (2) Reviewer #2 states that *"a helical edge state Dirac point ... is characterized by a vanishing LDOS"*. This is an incorrect statement. We think that the referee mistakenly starts here from the picture of a **two-dimensional Dirac cone**, like, e.g., in the 2D bulk states of graphene or the 2D surface states of a 3D TI. In this situation the DOS would indeed decrease linearly towards the Dirac point (V-shaped) and consequently show a dip there. It seems that she/he associates the zero bias anomaly seen in the tunneling spectrum of the free zigzag edge (which has a completely different origin, see below) with this dip. However, this contradicts the objective physical situation, because we are looking here at a **one-dimensional** edge state for which Reviewer #2's assertion is incorrect. This has been fully clarified by Reviewer #3 who we would like to quote here: *"For linearly dispersed edge states, the LDOS (dI/dV spectrum) should be a constant (dk/dE = constant) within single-particle picture, which is inconsistent with the "V" shape. In other words, it is impossible to determine Dirac point energy using dI/dV spectrum. So I don't think reviewer 2's "putative" assignment of Dirac point energy is reasonable"*.
- (3) In our previous response we have included the result of a band calculation showing that the DP is indeed located below the valence band maximum (and hence not at the Fermi level). This point was also explicitly addressed in the revised version of the paper. The referee now comments *"that model calculations can in many cases only provide a qualitative picture of the electronic structure."* We wish to point out that the reported band structure was not taken from some model calculation but rather is the result of a full scale *ab-initio* DFT calculation for the realistic atomic configuration of a bismuthene nano-ribbon on a SiC(0001) substrate, as documented in Ref. [21] of our paper. The calculated position of the DP is thus of fairly high reliability.
- (4) Reviewer #2 claims that her/his arguments *"are solely based on the experimental data presented by the authors in Fig. 3 and 4"* and *"wonder[s] which experimental data presented in the paper lead to their conclusion."* The latter question is precisely answered by Fig. 4a which clearly shows that the linear dispersion of the "electron pocket" extrapolates to a DP energy of approx. -0.4 at zero momentum (Gamma-point).

The central problem of this referee's view is the incorrect association of the zero bias anomaly (ZBA) in our tunneling spectra (seen in Fig. 3a) with a DP. However, the origin of the ZBA is of a quite different nature, as nicely summarized by Reviewer #3: *"On the other hand, the authors' prior work (Ref. [21] and [23]) provide substantial evidence that this feature is smoking-gun evidence of Tomonaga-Luttinger liquid (TLL) behavior, a celebrated strongly-correlated state of 1D system. In particular, Ref. [23] provides very detailed power-law scaling analysis that is in excellent agreement with the TLL picture."* This point has already been addressed in footnote [31] but we will shift this information now to the main text:

"The smooth and essentially featureless edge spectra confirm a homogeneous metallic LDOS closely confined to the zigzag edge, in full correspondence to the observations at armchair edges [21]. Note that the V-shaped dip at zero bias is attributed to Tomonaga-Luttinger liquid physics in the 1D edge state, based on its specific energy- and temperature dependence [23], and should not be confused with a gapped Dirac point."

Finally, concerning the remarks of Reviewers #1 and #3 on this specific criticism of Reviewer #2:

Reviewer #1:

I understand that the issue is on the Dirac-point energy. Although Referee #2 does not see, the QPI dispersions shown in Fig. 3h and Fig. 4a suggest that the bottom of the band responsible for the QPI, which should be close to the Dirac-point energy, is below the Fermi energy, as the authors state in the manuscript and the reply. Referee #2 pays more attention to the LDOS spectrum of the free zigzag edge that shows a V-shaped bottom near E_F . I agree with Referee #2 that this suggests the Dirac-point near E_F . Thus, I recommend the authors explain why the Dirac-point energies are different between the QPI dispersions and the tunneling spectrum.

We believe that our above reasoning should resolve this point: there is no difference in DP energies.

Reviewer #3:

In my opinion the authors' response in previous rebuttal letter is quite convincing. Yet, reviewer 2 doesn't seem to believe/understand this picture.

We thank this referee for clarifying the total picture.

Back to the remaining remarks of **Reviewer #2**:

Other major comment

I appreciate that the authors share my opinion. It is the breaking of bulk symmetries that lifts topological protection. While the coupling to other modes -- non-trivial or not -- can induce backscattering, it does not break or lift the topological protection of a quantum spin Hall insulator (=the bulk), as the authors claim in the title. Therefore, the manuscript's title and some of its content are misleading, if not physically incorrect, and should be amended "backscattering induced by edge-hybridization" or so would be more appropriate.

We have obviously a semantic disagreement on the term "topological protection". What Reviewer #2 seems to refer to is the "symmetry-related protection of topological order" which is indeed governed by **bulk** symmetries. Here, however, we explicitly address the topological protection of the extended edge states in a QSH insulator against single-particle backscattering, and hence against localization. It is in this sense that we use the phrase "topological protection", in agreement with a wide body of literature.

We fully agree that the inducement of edge state backscattering – by whatever mechanism – does not necessarily lift the topological character of the QSH insulator as a whole, as long as the relevant global symmetries stay preserved. Nonetheless, placing, e.g., a magnetic impurity on the edge of a QSH insulator (i.e., breaking time reversal symmetry (TRS) only locally) will open a channel for backscattering, without changing the QSH character of the bulk! In our specific case, we do not even break local TRS, but simply affect spin-momentum locking by pairwise edge-coupling, thereby demonstrating an alternative mechanism for opening backscattering channels. This is why we have called the effect "lifting of topological protection", referring again to the edge states, not the entire bulk.

In order to take the semantic edge out of this discussion we are willing to compromise and change the title to:

"Effective lifting of the topological protection of quantum spin Hall edge states by edge coupling"

Minor comment

While it is my understanding that the presentation of experimental data in their actual units is preferable, I do respect the authors' choice. Nevertheless, I insist on the labeling of the STM topography's apparent height at one position in the manuscript, because this could be a relevant piece of information for readers who are not yet familiar with the investigated material platform. Fig. 2a or d seem to be suitable for that purpose.

We follow this advice by labeling all topographic STM maps by their respective tip height scales.